# DECap: Towards Generalized Explicit Caption Editing via Diffusion Mechanism

## Abstract

Explicit Caption Editing (ECE) — refining reference image captions through a sequence of explicit edit operations (*e.g.*, KEEP, DETELE words) — has raised significant attention due to its explainable and human-like nature. After training with carefully designed reference and ground-truth caption pairs, state-of-the-art ECE models exhibit limited generalization ability beyond the original training data distribution, *i.e.*, they are tailored to refine content details only in *in-domain samples* but fail to correct errors in *out-of-domain samples*. To this end, we propose a new Diffusion-based Explicit Caption editing method: DECap. Specifically, we reformulate the ECE task as a denoising process under the diffusion mechanism, and introduce innovative edit-based noising and denoising processes. The noising process can help to eliminate the need for meticulous paired data selection by directly introducing word-level noises (*i.e.*, random words) for model training, learning diverse distribution over input reference captions. The denoising process involves the explicit predictions of edit operations and corresponding content words, refining reference captions through iterative step-wise editing. To further improve the inference speed for caption editing, DECap discards the prevalent multi-stage design and directly generates edit operations and content words simultaneously. Extensive experiments have demonstrated the strong generalization ability of DECap in various caption editing scenarios. More interestingly, it also shows great potential in improving both the quality and controllability of caption generation.

## 1 Introduction

Explicit Caption Editing (ECE), emerging as a novel task within the broader domain of caption generation, has raised increasing attention from multimodal learning community (Wang et al., 2022). As shown in Figure 1, given an image and a reference caption (Ref-Cap), ECE aims to explicitly predict a sequence of edit operations, which can translate the Ref-Cap to ground-truth caption (GT-Cap). Compared to conventional image captioning methods which generate captions from scratch (Vinyals et al., 2015; Xu et al., 2015; Chen et al., 2017; Anderson et al., 2018), ECE aims to enhance the quality of existing captions in a more explainable, efficient, and human-like manner.

Currently, existing ECE methods primarily rely on two prevalent benchmarks for model training and evaluation, *i.e.*, COCO-EE and Flickr30K-EE (Wang et al., 2022). Specifically, both datasets are carefully constructed to emphasize the refinement of content details while preserving the original caption structure. As shown in Figure 1, each ECE instance consists of an image along with a Ref-Cap (*e.g.*, two birds standing on a bench near the water) and a paired GT-Cap (man sitting on a bench overlooking the ocean). For this in-domain[1] sample, state-of-the-art ECE models can effectively improve the quality of the Ref-Cap. However, we found that existing ECE models have limited generalization ability when faced with out-of-domain samples. Take the model TIger (Wang et al., 2022) as an example, given a highly similar Ref-Cap with a single wrong word (man sitting on a bench running the ocean), it not only deletes the wrong word but also removes other accurate words. Meanwhile, when faced with more

---

[1]In this paper, we use the Levenshtein ratio (ratio) to quantify the similarity between two captions by considering their length and the edit distance needed to transform one into the other. The range of ratio is from 0 to 1, where a higher value indicates higher similarity. By "in-domain", we mean that the Ref-Cap and GT-Cap have a similar Levenshtein ratio as training samples. More details are left in the Appendix.

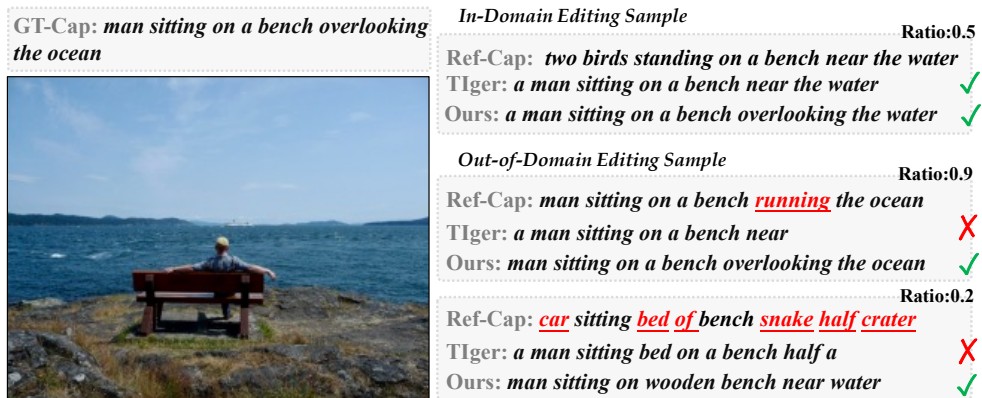

Figure 1: Editing results of ECE models. *In-domain* sample denotes that the Ref-Cap is from COCO-EE test set. *Out-of-domain* sample denotes the Ref-Caps are constructed by directly replacing the GT-Cap with random words. "Ratio" is the Levenshtenin ratio[1] between Ref-Cap and GT-Cap.

irrelevant Ref-Cap (`car` `sitting` `bed` `of` `bench` `snake` `half` `crater`), TIger fails to correct all errors or introduce sufficient accurate details. Obviously, this limited generalization ability will limit their utilization in real-world scenarios.

To address this limitation, we propose a novel diffusion-based ECE model, denoted as DECap, which reformulates the ECE task as a series of deonising process steps. Specifically, we design an edit-based noising process that constructs editing samples by introducing word-level noises (*i.e.*, random words) directly into the GT-Caps to obtain Ref-Caps. This noising process is parameterized by the distributions over both edit operations (*e.g.*, KEEP, DELETE, INSERT, and REPLACE) and caption lengths, which can not only avoid the meticulous selection of Ref-GT caption pairs but also help ECE models to learn a more adaptable distribution over Ref-Caps, capturing a broader spectrum of editing scenarios. Then, we train model DECap to refine Ref-Caps through an edit-based denoising process, which contains the iterative predictions of edit operations and content words. Meanwhile, our DECap discards the prevalent multi-stage architecture designs and directly generates edit operations and content words simultaneously, which can significantly accelerate the inference speed with simple Transformer encoder architectures. Extensive ablations have demonstrated that DECap can not only achieve outstanding editing performance on the challenging ECE benchmarks but also achieve competitive performance with conventional image captioning models by editing a series of random words. Furthermore, DECap even shows potential for word-level controllable captioning, which is beyond the ability of existing controllable captioning models (Cornia et al., 2019; Deng et al., 2020). In summary, *DECap realizes a strong generalization ability across various in-domain and out-of-domain editing scenarios, and showcases great potential in improving the quality and controllability of caption generation, keeping the strong explainable ability.*

In summary, we make several contributions in this paper: 1) To the best of our knowledge, we are the first work to point out the poor generalization issues of existing ECE models. 2) DECap is the first diffusion-based model, which pioneers the use of the discrete diffusion mechanism for ECE. 3) DECap shows strong generalization ability across various editing scenarios, achieving outstanding performance on ECE benchmarks and remarkable results in even editing random word sequences. 4) Thanks to our designs, DECap has a much faster inference speed than existing ECE methods.

## 2 RELATED WORK

**Explicit Caption Editing (ECE).** Given the image, ECE aims to refine existing Ref-Caps through a sequence of edit operations, which was first proposed by Wang *et.al.* (Wang et al., 2022). Specifically, by realizing refinement under the explicit traceable editing path composed of different edit operations (*e.g.*, KEEP/DELETE/ADD), this task encourages models to enhance the caption quality in a more explainable and efficient manner. However, existing ECE benchmarks are carefully designed, targeting on the refinement of specific content details, which leads to a limited model generalization ability across diverse real-world editing scenarios beyond the training data distribu-

tion. Meanwhile, existing editing models (Mallinson et al., 2020; Wang et al., 2022; Reid & Neubig, 2022) tend to perform the editing with multiple sub-modules sequentially. For example, conducting the insertion operation by first predicting the ADD operation, then applying another module to predict the specific word that needs to be added. In this paper, we construct Ref-Caps by directly noising the GT-Caps at word-level through a novel edit-based noising process, allowing the model to capture various editing scenarios during training. We further optimize the model architecture to predict both edit operations and content words parallelly, which can significantly accelerate the editing speed.

**Diffusion-based Captioning Models.** Taking inspiration from the remarkable achievements of diffusion models in image generation (Austin et al., 2021; Rombach et al., 2022), several pioneering works have applied the diffusion mechanism for caption generation. To the best of our knowledge, existing diffusion-based captioning works can be mainly categorized into two types: 1) **Continuous Diffusion**: They aim to convert discrete words into continuous vectors (*e.g.*, word embeddings (He et al., 2023) and binary bits (Chen et al., 2022; Luo et al., 2023)) and apply the diffusion process with Gaussian noises. 2) **Discrete Diffusion**: They aim to extend the diffusion process to discrete state spaces by directly noising and denoising sentences at the token level, such as gradually replacing tokens in the caption with a specific [MASK] token and treating the denoising process as a multi-step mask prediction task starting from an all [MASK] sequence (Zhu et al., 2022). As the first diffusion-based ECE model, in contrast to iterative mask replacement, which only trains the ability to predict texts for [MASK] tokens, our edit-based noising and denoising process can help our model to learn a more flexible way of editing (*e.g.*, insertion, deletion, and replacement) by different edit operations. Meanwhile, our model shows its great potential in editing random word sequences to generate captions with competitive performance compared to diffusion-based captioning models.

# 3 DECap: Diffusion-based Explicit Caption Editing

In this section, we first give a brief introduction of the task formulation of ECE and the preliminaries about the discrete diffusion mechanism in Sec. 3.1. Then, we show the edit-based noising and denoising process in Sec. 3.2. We introduce our Transformer-based model architecture in Sec. 3.3. Lastly, we demonstrate the details of training objectives and inference process in Sec. 3.4.

## 3.1 Task Formulation and Preliminaries

**Explicit Caption Editing.** Given an image $I$ and a reference caption (Ref-Cap) $\boldsymbol{x}^r = \{w_r^1, ..., w_r^n\}$ with $n$ words, ECE aims to explicitly predict a sequence of $m$ edit operations $E = \{e^1, ..., e^m\}$ to translate the Ref-Cap close to the ground-truth caption (GT-Cap) $\boldsymbol{x}_0 = \{w_0^1, ..., w_0^k\}$ with $k$ words.

**Edit Operations.** Normally, different ECE models may utilize different basic edit operations, and they mainly focus on the reservation (*e.g.*, KEEP) and deletion (*e.g.*, DELETE) of existing contents, and the insertion (*e.g.*, ADD, INSERT) of new contents. Without loss of generality, in this paper, we utilize the four Levenshtein edit operations for both the noising and denoising process, including: 1) KEEP, the keep operation preserves the current word unchanged; 2) DELETE, the deletion operation removes the current word; 3) INSERT, the insertion operation adds a new word after the current word; 4) REPLACE, the replacement operation overwrites the current word with a new word.

**Discrete Diffusion Mechanism.** For diffusion models in the discrete state spaces for text generation, each word of sentence $\boldsymbol{x}_t$ is a discrete random variable with $K$ categories, where $K$ is the word vocabulary size. Denoting $\boldsymbol{x}_t$ as a stack of one-hot vectors, the noising process can be written as:

$$q(\boldsymbol{x}_t|\boldsymbol{x}_{t-1}) = Cat(\boldsymbol{x}_t; p = \boldsymbol{x}_{t-1}\boldsymbol{Q}_t),\qquad(1)$$

where $Cat(\cdot)$ is a categorical distribution and $\boldsymbol{Q}_t$ is a transition matrix applied to each word in the sentence independently: $[\boldsymbol{Q}_t]_{i,j} = q(w_t = j|w_{t-1} = i)$. Existing discrete diffusion text generation works (Austin et al., 2021; He et al., 2022; Zhu et al., 2022) mainly follow the noising strategy of BERT (Devlin et al., 2018), where each word stays unchanged or has some probability transitions to the [MASK] token or other random words from the vocabulary. Meanwhile, they incorporate an absorbing state for their diffusion model as the [MASK] token:

$$[\boldsymbol{Q}_t]_{i,j} = \begin{cases} 1 & \text{if } i = j = \texttt{[MASK]}, \\ \beta_t & \text{if } j = \texttt{[MASK]}, i \neq \texttt{[MASK]}, \\ 1 - \beta_t & \text{if } i = j \neq \texttt{[MASK]}. \end{cases}\qquad(2)$$

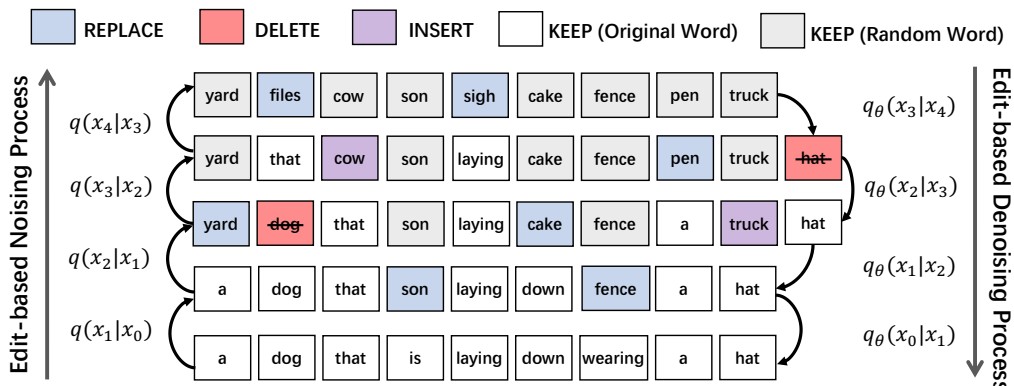

Figure 2: Edit-based noising process for DECap. **Blue** represents the REPLACE operation, **red** represents the DELETE operation, **purple** represents the INSERT operation, **white** and **grey** represent the KEEP operation for original word and random word respectively.

After a sufficient number of noising steps, this Markov process converges to a stationary distribution $q(\boldsymbol{x}_T)$ where all words are replaced by the [MASK] token. Discrete diffusion works then train their models to predict target words for [MASK] tokens as the denoise process $p_\theta(\boldsymbol{x}_{t-1}|\boldsymbol{x}_t, t)$, and generate the sentence by performing a series of denoising steps from an all [MASK] token sequence:

$$P_\theta(\boldsymbol{x}_0) = \prod_{t=1}^{T} p_\theta(\boldsymbol{x}_{t-1}|\boldsymbol{x}_t, t). \tag{3}$$

## 3.2 DISCRETE DIFFUSION FOR ECE

Taking inspiration from the discrete process where a noised sentence is iteratively refined into the target sentence, we reformulate the ECE with a discrete diffusion mechanism and parameterize the noising and denoising process by way of sampled discrete edit operations applied over the caption words. *This edit-based noising and denoising process can successfully mitigate the need for paired Ref-GT caption pairs, as we only need to conduct the diffusion process on the original GT-Cap.*

**Edit-based Noising Process.** Different from directly transiting one word to another, the edit-based noising process gradually adds word-level noises to the caption $\boldsymbol{x}_{t-1}$ based on different edit operations. For any time step $t \in (0, T]$, the edit-based noising process is defined as

$$q(\boldsymbol{x}_t|\boldsymbol{x}_{t-1}) = p(\boldsymbol{x}_t|\boldsymbol{x}_{t-1}, E_t^N) \cdot Cat(E_t^N; p = \boldsymbol{x}_{t-1}\boldsymbol{Q}_t), \tag{4}$$

where $Cat(\cdot)$ is a categorical distribution and $\boldsymbol{Q}_t$ here is a transition matrix assigning edit operation for each word in the caption $\boldsymbol{x}_{t-1}$ independently: $[\boldsymbol{Q}_t]_{i,j} = q(e_t = j|w_{t-1} = i)$. Subsequently, $E_t^N = \{e_t^1, e_t^2, ..., e_t^l\}$ is a sequence of noising edit operations which has the same length with the caption $\boldsymbol{x}_{t-1} = \{w_{t-1}^1, w_{t-1}^2, ..., w_{t-1}^l\}$[2], where each edit operation $e_t^i$ is operated on the corresponding word $w_{t-1}^i$ to get $\boldsymbol{x}_t$. Specifically, $\boldsymbol{Q}_t$ is parameterized by the distribution over both edit types and the GT-Cap length $k$ with an absorbing state as the random word[3].

$$[\boldsymbol{Q}_t]_{i,j} = \begin{cases} 1 & \text{if } j = \text{KEEP}, \quad i = \text{random word}, \\ \alpha_t^k & \text{if } j = \text{REPLACE}, i \neq \text{random word}, \\ \beta_t^k & \text{if } j = \text{DELETE}, \quad i \neq \text{random word}, \\ \gamma_t^k & \text{if } j = \text{INSERT}, \quad i \neq \text{random word}, \\ 1 - \alpha_t^k - \beta_t^k - \gamma_t^k, & \text{if } j = \text{KEEP}, \quad i \neq \text{random word}. \end{cases} \tag{5}$$

Subsequently, as the example shown in Figure 2, being operated with $e_t$, each word $w_{t-1}$ has a probability of $\alpha_t^k$ to be replaced by another random word, has a probability of $\beta_t^k$ to be removed from the caption, and has a probability of $\gamma_t^k$ to be added with a random word after it, leaving the probability of $\delta_t^k = 1 - \alpha_t^k - \beta_t^k - \gamma_t^k$ to be unchanged. Accordingly, the distribution over the GT-Cap length $k$ can ensure a smooth increase of noised words for each noising step from $\boldsymbol{x}_0$ to $\boldsymbol{x}_T$.

---

[2]Generally, the length of captions may vary in different steps. For simplicity, we slightly use $l$ to denote the length of all other $\boldsymbol{x}_t$ captions in this paper.

[3]In this paper, "random word" refers to word from the vocabulary except those in the original caption.

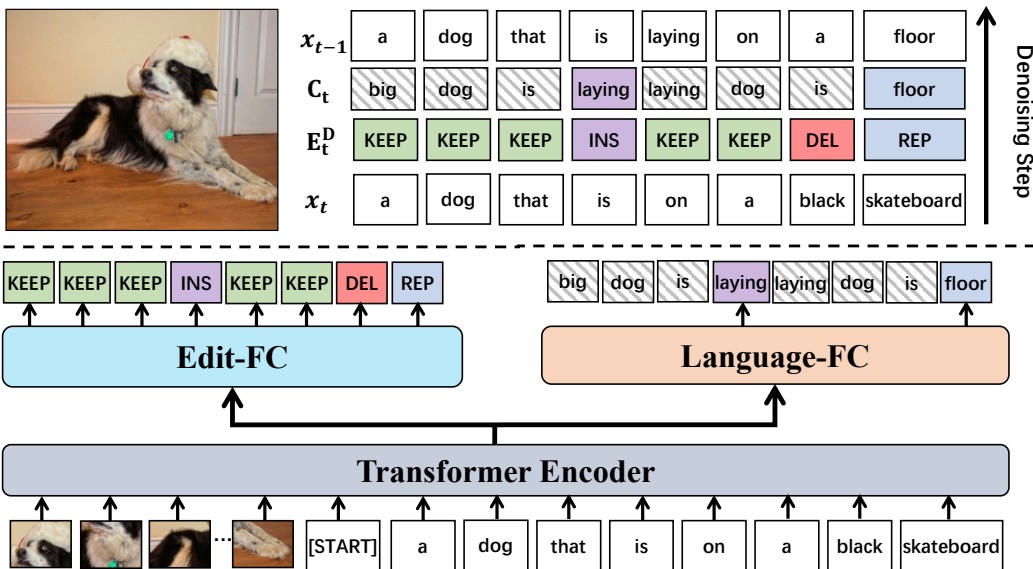

Figure 3: The edit-based denoising step and architecture of DECap. DECap will predict a sequence of edit operations and content words to transform the caption. Specifically, the contents words are used only when the predicted corresponding edit operation is INSERT or REPLACE, while the rest of the predicted words are abandoned, *i.e.*, the shaded words.

The distribution over edit types ensures the balance between different noising operations and the learning of different denoising abilities: 1) To learn the ability to INSERT new words, we remove words by DELETE operation. 2) To learn the ability to DELETE incorrect words, we add random words by INSERT operation. 3) To learn the ability to directly REPLACE incorrect words, we change current words into random words by REPLACE operation. 4) To learn the ability to KEEP correct content, we leave the correct words unchanged by KEEP operation. Meanwhile, if the word has already been noised into the random word, it will not be re-noised again. Then through a sufficient number of noising steps $T$, the caption will be noised into a random word sequence.

**Edit-based Denoising Process.** The edit-based denoising process aims to iteratively edit $\boldsymbol{x}_T$ to $\boldsymbol{x}_0$ by predicting appropriate edit operations. Specifically, given the image $I$ and the caption $\boldsymbol{x}_t = \{w_t^1, w_t^2, ..., w_t^l\}$, we model this edit-based denoising process with the explicit prediction of both edit operations and content words which transform $\boldsymbol{x}_t$ to $\boldsymbol{x}_{t-1}$:

$$p_\theta(\boldsymbol{x}_{t-1}|\boldsymbol{x}_t, t, I) = p(\boldsymbol{x}_{t-1}|\boldsymbol{x}_t, E_t^D, C_t) \cdot p(E_t^D, C_t|\boldsymbol{x}_t, t, I), \quad (6)$$

where $p_\theta$ parameterized the model to predict a sequence of denoising edit operations $E_t^D = \{e_t^1, e_t^2, ..., e_t^l\}$, together with a sequence of content words $C_t = \{c_t^1, c_t^2, ..., c_t^l\}$ which all have the same length with $\boldsymbol{x}_t$. As the example shown in Figure 3, the denoising step transforms the caption $\boldsymbol{x}_t$ to $\boldsymbol{x}_{t-1}$ based on the edit operations and predicted words, *i.e.*, for each word $w_t^i$, we keep the original word if it is predicted operation $e_t^i$ is KEEP, remove the word if it is predicted operation $e_t^i$ is DELETE, copy the original word and add a new word $c_t^i$ after it if predicted operation $e_t^i$ is INSERT, and replace it with a new word $c_t^i$ if predicted operation $e_t^i$ is REPLACE. We then feed the output of this step into the model and perform the next denoising step. Following this, we can generate the caption by performing a series of denoising steps from an all random word sequence:

$$P_\theta(\boldsymbol{x}_0) = \prod_{t=1}^{T} p_\theta(\boldsymbol{x}_{t-1}|\boldsymbol{x}_t, t, I). \quad (7)$$

### 3.3 TRANSFORMER-BASED MODEL ARCHITECTURE

The DECap is built based on the standard Transformer (Vaswani et al., 2017) architecture, which has strong representation encoding abilities. To facilitate the denoising process, we further construct DECap with a parallelized system for efficient generation of both edit operations and content words.

**Feature Extraction.** Given the image $I$ and caption $x_t$, we construct the input for the model as a sequence of visual tokens and word tokens. Specifically, we encode the image $I$ into visual tokens through pre-trained visual backbones such as CLIP or Faster R-CNN. The word tokens are represented by the sum of word embedding, position encoding, and segment encoding. Meanwhile, following previous works (Ho et al., 2020; Austin et al., 2021; Li et al., 2022), we encode the time step $t$ as a sinusoidal embedding the same way as the position encoding, adding it to the word tokens.

**Model Architecture.** As shown in Figure 3, given the visual-word token sequence with a special connecting token, *e.g.*, [START], we first utilize the Transformer encoder blocks with self-attention and co-attention layers to learn the multi-modal representations of each token. We then use two simple yet effective FC layers to predict the edit operation and content word for each word token. Specifically, by feeding the hidden states of word tokens as input, 1) the Edit-FC generates the edit operation sequence $E_t^D$ by making a four-category classification for each word, *i.e.*, $e_t \in \{\text{REPLACE}, \text{DELETE}, \text{INSERT}, \text{KEEP}\}$. 2) In parallel, the Language-FC maps each hidden state to a distribution over the vocabulary to predict specific words to generate the content word sequence $C_t$. Following the denoising step in Sec. 3.2, we then transform the caption $x_t$ to $x_{t-1}$ based on the edit operations and content words for the next step.

### 3.4 TRAINING OBJECTIVES AND INFERENCE

**Training.** Following previous discrete diffusion works (Austin et al., 2021; Zhu et al., 2022), we train the model to directly predict the original ground-truth caption $x_0$ for caption $x_t$:

$$\mathcal{L} = \mathcal{L}_{Edit} + \mathcal{L}_{Languge} = -\log p_\theta(E_t^G|\boldsymbol{x}_t, t, I) + -\log p_\theta(C_t^G|\boldsymbol{x}_t, t, I), \tag{8}$$

where $E_t^G$ and $C_t^G$ are ground truth edit operations and content words constructed based on the $x_0$ and $x_t$. And $\mathcal{L}_{Edit}$ and $\mathcal{L}_{Languge}$ are typically cross-entropy loss over the distribution of predicted edit operations and content words. Specifically, the $\mathcal{L}_{Languge}$ part is only trained to predict content words for the input words assigned with INSERT and REPLACE operations.

**Inference.** Given the image and caption $\boldsymbol{x}_t$, *i.e.*, $t \in (t, T]$, the diffusion model predicts $\boldsymbol{x}_{t-1}$, $\boldsymbol{x}_{t-2}$ iteratively for $t$ denoising steps, and produces the final result of $\boldsymbol{x}_0$.

## 4 EXPERIMENTS

### 4.1 EXPERIMENTAL SETUP

**Datasets.** We evaluated our DECap on both popular ECE and image captioning benchmarks, *i.e.*, COCO-EE, Flickr30K-EE[4] (Wang et al., 2022) and COCO (Lin et al., 2014) dataset. Specifically, the COCO-EE contains 97,567 training instances, 5,628 validation instances, and 5,366 testing instances, where each editing instance consists of one image and one corresponding Ref-GT caption pair. The COCO dataset contains 123,287 images with 5 human-annotated captions for each. In this paper, we utilized the widely adopted Karpathy splits (Karpathy & Fei-Fei, 2015), which contain 113,287 training images, 5,000 validation images, and 5,000 test images.

**Evaluation Metrics.** For the quality evaluation, we utilized all the prevalent accuracy-based metrics following prior works, which include BLEU-N (Papineni et al., 2002), METEOR (Banerjee & Lavie, 2005), ROUGE-L (Lin, 2004), CIDEr-D (Vedantam et al., 2015), and SPICE (Anderson et al., 2016). Meanwhile, we also computed the inference time to evaluate the model efficiency.

### 4.2 GENERALIZATION ABILITY IN EXPLICIT CAPTION EDITING

In this subsection, we evaluated the generalization ability of our model with both in-domain and out-of-domain evaluation on the Flickr30K-EE benchmark. We trained both TIger and DECap with the same word vocabulary sized 12,071. Specifically, during training, TIger used the complete editing instance (*i.e.*, the image and the Ref-GT caption pair) while DECap only used the image and the GT caption with diffusion step $T = 10$.

---

[4]Due to the limited space, more details are left in the Appendix.

| Model | Unpaired Data | Step | Quality Evaluation | | | | | | | Inference Time(ms) |
|---|---|---|---|---|---|---|---|---|---|---|
| | | | B-1 | B-2 | B-3 | B-4 | R | C | S | |
| Ref-Caps | — | — | 50.0 | 37.1 | 27.7 | 19.5 | 48.2 | 129.9 | 18.9 | — |
| TIger | ✗ | 4 | 50.3 | 38.5 | 29.4 | 22.3 | 53.1 | 176.7 | 31.4 | 614.23 |
| **DECap** | ✔ | 4 | 55.5 | 41.7 | 31.5 | 23.3 | 52.7 | 173.7 | 29.8 | 277.30 |
| **DECap** | ✔ | 5 | 56.0 | **42.0** | **31.6** | **23.5** | 53.0 | 176.2 | 31.4 | 335.45 |
| **DECap** | ✔ | 6 | **56.1** | 41.9 | 31.4 | 23.4 | **53.1** | **177.0** | **32.2** | 409.99 |

Table 1: The "in-domain" evaluation of our model and state-of-the-art ECE model on the COCO-EE test set. "Ref-Caps" denotes the initial quality of given reference captions.

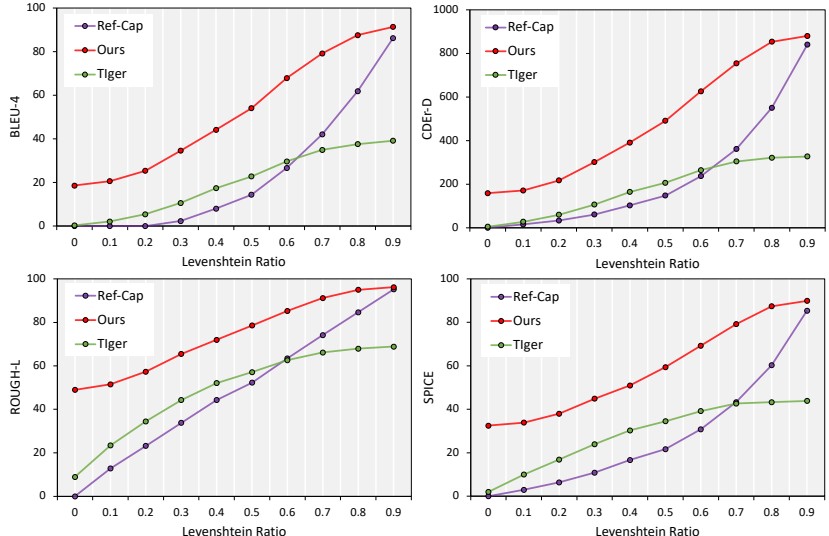

Figure 4: Performance of our model and state-of-the-art ECE models on out-of-domain GT-based reference captions. "Ref-Caps" denotes the quality of given GT-based reference captions.

### 4.2.1 IN-DOMAIN EVALUATION

**Settings.** Since the COCO-EE dataset was carefully designed to emphasize the refinement of content details, its training set and test set have similar distribution (*i.e.*, Ratio[1] around 0.5 for most instances), thus we directly compared the performance of each model on the COCO-EE test set as the in-domain evaluation. We evaluated edited captions against their single GT-Cap.

**Results.** The in-domain evaluation results are reported in Table 1. From the table, we can observe: 1) For the quality evaluation, TIger achieves its best performance using four editing steps, while our DECap achieves competitive results with the same step (*e.g.*, better BLEU scores but slightly lower CIDEr-D score). With more editing steps, DECap can further improve the quality of captions, outperforming TIger on all metrics. *It is worth noting that TIger was even trained on the in-domain Ref-GT caption pairs.* 2) For the efficiency evaluation, DECap achieves significantly faster inference speed than TIger even with more editing steps. This is because that DeCap predicts edit operations and content words simultaneously but TIger needs to conduct editing by three sequential modules.

### 4.2.2 OUT-OF-DOMAIN EVALUATION

In addition to editing the carefully designed reference captions, we sought to evaluate ECE models on a broader spectrum of editing scenarios that deviate from the in-domain training data. Specifically, we further evaluated DECap on two types of out-of-domain samples:

**Setting A: GT-based Reference Captions.** The GT-based reference captions were constructed based on the GT-Caps in the COCO-EE test set. We systematically replaced words in the GT-Caps with other random words from the vocabulary, resulting in the creation of various out-of-domain Ref-Caps. These Ref-Caps varied in terms of their Levenshtein ratio, ranging from 0.9 (*i.e.*, with only a few incorrect words) to 0 (*i.e.*, where all words were replaced with random ones). We evaluated edited captions against their single GT-Cap.

| Model | Unpaired Data | Step | Quality Evaluation | | | | | | | Inference Time(ms) |
|---|---|---|---|---|---|---|---|---|---|---|
| | | | B-1 | B-2 | B-3 | B-4 | R | C | S | |
| TIger | ✗ | 10 | 14.7 | 4.6 | 1.9 | 0.9 | 13.5 | 3.0 | 1.2 | 1413.16 |
| **DECap** | ✔ | 10 | **74.7** | **57.4** | **42.1** | **30.0** | **55.3** | **102.5** | **19.6** | 684.32 |

Table 2: Performance of our model and state-of-the-art ECE model on Zero-GT reference captions.

| | Model | B-1 | B-2 | B-3 | B-4 | M | R | C | S |
|---|---|---|---|---|---|---|---|---|---|
| Autoregress. | Up-Down (Anderson et al., 2018) | 77.2 | — | — | 36.2 | 27.0 | 56.4 | 113.5 | 20.3 |
| | Transformer (Sharma et al., 2018) | 76.1 | 59.9 | 45.2 | 34.0 | 27.6 | 56.2 | 113.3 | 21.0 |
| | LBPF (Qin et al., 2019) | 77.8 | — | — | 37.4 | 28.1 | 57.5 | 116.4 | 21.2 |
| | SAGE (Yang et al., 2019) | 77.6 | — | — | 36.9 | 27.7 | 57.2 | 116.7 | 20.9 |
| | AoANet (Huang et al., 2019) | 77.4 | — | — | 37.2 | 28.4 | 57.5 | 119.8 | 21.3 |
| Non-Autoregress. | MNIC (Gao et al., 2019) | 75.4 | 57.7 | 42.6 | 30.9 | 27.5 | 55.6 | 108.1 | 21.0 |
| | CMAL (Guo et al., 2020) | 78.5 | — | — | 35.3 | 27.3 | 56.9 | 115.5 | 20.8 |
| | SATIC (Zhou et al., 2021) | 77.3 | — | — | 32.9 | 27.0 | — | 110.0 | 20.5 |
| | *Diffusion-based Method* | | | | | | | | |
| | Bit Diffusion (Chen et al., 2022) | — | — | — | 34.7 | — | 58.0 | 115.0 | — |
| | SCD-Net (Luo et al., 2023) | **79.0** | **63.4** | **49.1** | **37.3** | 28.1 | 58.0 | 118.0 | 21.6 |
| | DDCap (Zhu et al., 2022) | — | — | — | 35.0 | 28.2 | 57.4 | 117.8 | 21.7 |
| | **DECap (step=10)** | 78.0 | 61.4 | 46.4 | 34.5 | 28.6 | 58.0 | 119.0 | 21.9 |
| | **DECap (step=15)** | 78.6 | 62.2 | 47.4 | 35.3 | **29.0** | **58.4** | **121.2** | **22.7** |

Table 3: Performance of DECap and state-of-the-art captioning models on COCO. The **best** and second best results are denoted with corresponding formats.

**Results.** As shown in Figure 4: Our model successfully improves the quality of all the GT-based Ref captions. In contrast, TIger struggles when editing Ref captions with either "minor" or "severe" errors, and even degrading the captions' quality (*e.g.*, Ref-Caps with a ratio larger than 0.6[4]) by inadvertently removing accurate words or failing to introduce accurate details.

**Setting B: Zero-GT Reference Captions.** To further evaluate the models' generalization ability on the content generation without utilizing any GT captions, we constructed Zero-GT reference captions based on the COCO test set. Specifically, each editing instance consists of a single image and a Ref-Cap with ten random words. Subsequently, we evaluated the edited captions against their corresponding five GT-Caps. All results are reported in Table 2.

**Results.** From Table 2 we can observe: 1) Given the image, both models achieve their best performance with ten editing steps, our DECap successfully edits the sentence with all random words into a coherent caption. In contrast, TIger faces challenges in doing so. 2) In terms of efficiency metrics, our DECap achieves significantly faster inference speed compared to TIger.

## 4.3 CAPTION GENERATING ABILITY

Surprised by the results of editing entirely random word sequences, we further investigated DECap's capacity for generating image captions when compared to existing image captioning methods.

**Settings.** We compared DECap with both state-of-the-art autoregressive and non-autoregressive approaches on COCO dataset. In particular, we focus on the comparison with the discrete diffusion-based captioning model DDCap (Zhu et al., 2022). DDCap introduces token-level noises into captions by gradually replacing tokens with [MASK] tokens. Subsequently, it generates captions starting from sequences filled with these [MASK] tokens. A transformer-based model is trained to predict specific content for [MASK] tokens. We trained DECap on the COCO training set with a vocabulary size of 23,531 together with different diffusion steps $T = 10$ and $T = 15$. During testing, we constructed input instances consisting of a single image from the COCO test set and a Ref-Cap with ten random words. The edited captions were then evaluated against their corresponding five GT-Caps.

**Results.** From Table 3 we can observe: 1) Within ten editing steps, DECap achieves competitive performance with SOTA autoregressive methods (*e.g.*, 119.0 vs. 119.8 in AoANet on CIDEr-D) and notably surpasses all conventional non-autoregressive methods. This to some extent highlights the ability of our edit-based method to mitigate the limitations of non-autoregressive methods, such as word repetition or omission issues. 2) When compared with diffusion-based captioning works, DE-

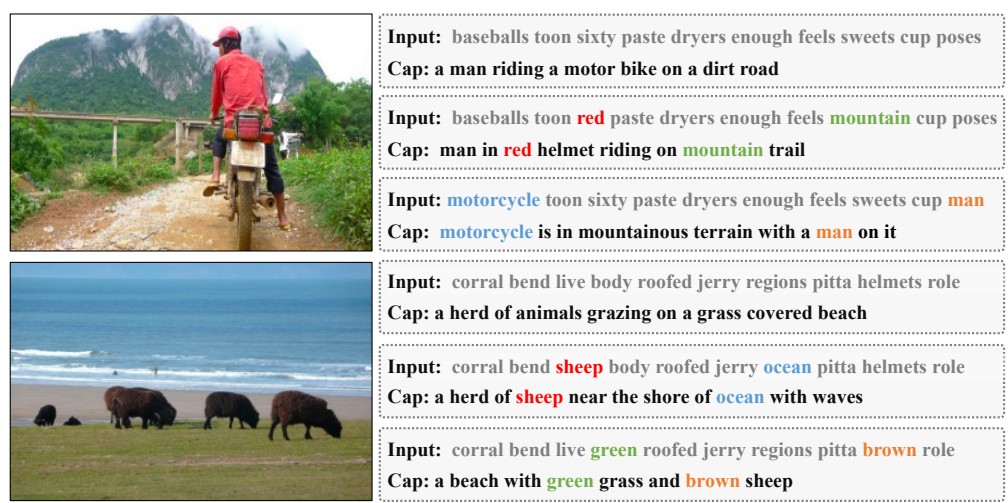

Figure 5: Controllability of DECap. The grey words represent random words from the vocabulary, and other colored words represent the manually placed control words.

Cap achieves superior performance on key metrics (*e.g.*, CIDEr-D and SPICE). While it falls slightly behind SCD-Net on BLEU, it's important to note that CIDEr and SPICE metrics are specifically designed for captioning evaluation and are better aligned with human judgments (than BLEU-N). 3) DECap achieves a significantly faster inference speed compared with another discrete diffusion-based model DDCap (675.80 vs. 3282.58ms). 4) Additionally, DECap can further boost the performance with more editing steps (*e.g.*, 121.2 on CIDEr-D with 15 steps) and keep the inference speed at a reasonable level (*i.e.*, 933.10ms). These results suggest the remarkable potential of DECap in improving the quality of caption generation in a more explainable and efficient edit-based manner.

### 4.4 POTENTIAL ABILITY: CONTROLLABLE IMAGE CAPTIONING (CIC)

Building on the remarkable generalization ability exhibited by DECap in both caption editing and generation, we further conducted a preliminary exploration of its potential in terms of controllability. Compared to existing CIC methods, which offer only coarse control over contents and structures, we can achieve precise and explicit control over caption generation through predefined control words.

**Settings.** We constructed input instances consisting of a single image from the COCO test set and a sentence with ten random words. Then, we replaced several random words with specific control words (*e.g.*, objects and attributes) at predefined positions based on the visual information of images.

**Results.** As shown in Figure 5, DECap is capable of editing sentences based on input control words, *i.e.*, all generated captions follow the order of the given control words with guaranteed fluency. Meanwhile, DECap shows its reasoning ability to generate relevant semantic content based on the control words: 1) Given the attributes (*e.g.*, color), DECap can generate specific contents with these attributes (*e.g.*, "red" → "helmet", "green" → "grass" and "brown" → "sheep"). 2) Given objects, DECap can generate further descriptions or related objects (*e.g.*, "mountain" → "trail" and "ocean" → "wave"). These results indicate the potential of DECap to enhance controllability and diversity, achieving a more direct and word-level control beyond existing CIC methods.

## 5 CONCLUSION

In this paper, we pointed out the challenge of limited generalization ability in existing ECE models. To this end, we proposed a novel diffusion-based ECE model, DECap, which reformulates ECE with a discrete diffusion mechanism, incorporating an innovative edit-based noising and denoising process. Extensive experiments have demonstrated DECap's strong generalization ability and potential for bridging the gap between caption editing and generation. Moving forward, we are going to: 1) extend our method into other modalities beyond images, *e.g.*., video captioning; 2) delve deeper into advanced techniques for finer controllability of DECap's editing process.

**Ethics Statement.** Our proposed ECE model may face the same potential ethical concerns as other existing ECE or image captioning works, such as suffering from severe bias issues (*e.g.*, gender bias (Hendricks et al., 2018)). Additionally, our method may also be maliciously utilized by using some improper control words, such as sensitive attributes. Apart from these general issues that already exist in the ECE or image captioning tasks, our paper has no additional ethical issues.

**Reproducibility Statement.** DECap is mainly implemented based on the released code of ViL-BERT (Lu et al., 2019) and evaluated on the publicly available datasets, including the ECE benchmarks (Wang et al., 2022) and COCO (Lin et al., 2014) dataset. We will also release all source codes and pretrained models.

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

## APPENDIX

The Appendix is organized as follows:

- In Sec. A, we show more details about the Levenshtein ratio.
- In Sec. B, we show the data distribution of the COCO-EE based on the Levenshtein ratio.
- In Sec. C, we show the implementation details.
- In Sec. D, we show more results about the generalization ability of DECap on the Fickr30K-EE dataset.
- In Sec. E, we provide the ablation study about the number of random words in caption generation.
- In Sec. F, we provide the ablation study about the distribution of edit types.
- In Sec. G, we provide more visualization results.

## A  DETAILS FOR LEVENSHTEIN RATIO

In this paper, we used the Levenshtein ratio to quantify the similarity between two captions by considering their length and the edit distance needed to transform one into the other. Specifically, for two captions with length $m$ and $n$, the Levenshtein ratio is calculated as:

$$\texttt{ratio} = \frac{m + n - ldist}{m + n} \tag{9}$$

where $ldist$ is the weighted edit distance based on the standard Levenshtein distance (Levenshtein et al., 1966). The Levenshtein distance refers to the minimum number of edit operations required to transform one sentence into another, including three Levenshtein operations `REPLACE`, `INSERT`, and `DELETE`. In the case of the weighted version, when calculating $ldist$, both `INSERT` and `DELETE` operations are still counted as $+1$, while each `REPLACE` operation incurs a cost of $+2$:

$$ldist = \mathbf{Num}(\texttt{INSERT}) + \mathbf{Num}(\texttt{DELETE}) + 2 * \mathbf{Num}(\texttt{REPLACE}) \tag{10}$$

where $\mathbf{Num}(\cdot)$ represents the number of different edit operations. The range of Levenshtein ratio yields from 0 to 1, where a higher value indicates higher similarity.

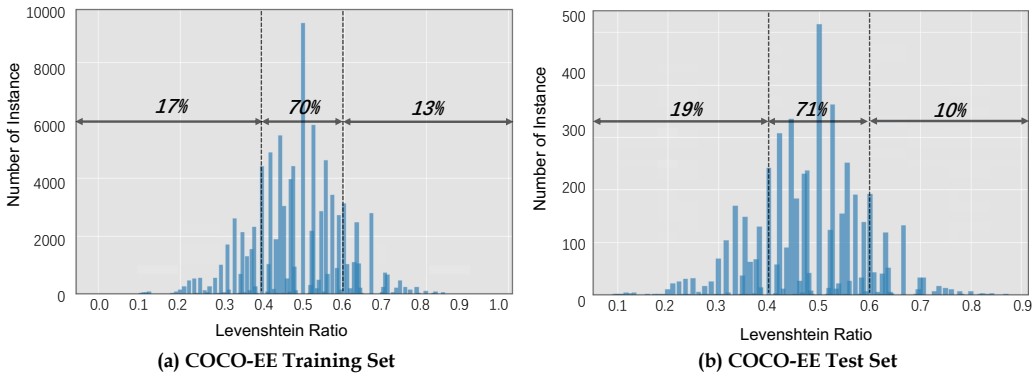

Figure 6: The data distribution of the COCO-EE dataset.

## B    DATA DISTRIBUTION OF COCO-EE

As illustrated in Fig 6, we can observe that the training set and test set of COCO-EE have similar distribution: 1) More than 70% of the editing instances have ratios ranging from 0.4 to 0.6, with the majority of them concentrated around 0.5. 2) There are very few samples with ratios below 0.4 or above 0.6, and almost no samples with ratios around 0.1 or 0.9. 3) The distribution of COCO-EE is highly uneven.

## C    IMPLEMENTATION DETAILS.

For image features, we used the ViT features extracted by the ViT-B/16 (Dosovitskiy et al., 2020) backbone from the pretrained CLIP model (Radford et al., 2021) with image patch size 16. For the edit-based noising process, we set $\alpha > \beta = \gamma$ to emphasize the denoising ability of replacement. For our diffusion model, we used the 12-layer Transformer encoder. We trained our model with Adam optimizer for 50 epochs, and we used a linear decay learning rate schedule with warm up. The initial learning rate was set to 1e-4. The inference time was evaluated as the average run time for each instance on a single A100 GPU with a mini-batch size of 1.

## D    MORE RESULTS ABOUT DECAP'S GENERALIZATION ABILITY

In this subsection, we evaluated the generalization ability of our model with both in-domain and out-of-domain evaluation on another ECE benchmark, *i.e.*, the Flickr30K-EE (Wang et al., 2022). Specifically, the Flickr30K-EE contains 108,238 training instances, 4,898 validation instances, and 4,910 testing instances, where each editing instance consists of one image and one corresponding Ref-GT caption pair. We trained both DECap and TIger on the Flickr30K-EE training set using the same word vocabulary sized 19,124. Specifically, during training, TIger used the complete editing instance (*i.e.*, the image and the Ref-GT caption pair) while DECap only used the image and the GT caption with diffusion step $T = 6$.

***In-Domain Evaluation***

**Settings.** We compare the performance of each model on the Flickr30K-EE test set as the in-domain evaluation. And we evaluated edited captions against their single ground-truth caption.

**Results.** The in-domain evaluation results are reported in Table 4. From the table, we can observe: 1) For the quality evaluation, TIger achieves its best performance using three editing steps, and our DECap achieves better results with the same step. With more editing steps, DECap can further improve the quality of captions on all metrics. *It is worth noting that TIger was even trained on the in-domain Ref-GT caption pairs*. 2) For the efficiency evaluation, DECap achieves significantly faster inference speed than TIger even with more editing steps. This is because that DeCap predicts edit operations and content words simultaneously but TIger needs to conduct editing by three sequential modules.

***Out-of-Domain Evaluation***

| Model | Unpaired Data | Step | Quality Evaluation | | | | | | | Inference Time(ms) |
|---|---|---|---|---|---|---|---|---|---|---|
| | | | B-1 | B-2 | B-3 | B-4 | R | C | S | |
| Ref-Caps | — | — | 34.7 | 24.0 | 16.8 | 10.9 | 36.9 | 91.3 | 23.4 | — |
| TIger | ✗ | 3 | 31.9 | 23.9 | 18.1 | 12.4 | 40.6 | 131.8 | 30.8 | 501.42 |
| **DECap** | ✔ | 3 | 37.6 | 27.5 | 19.8 | 13.7 | 40.8 | 134.0 | 31.0 | 214.46 |
| **DECap** | ✔ | 4 | **38.2** | **27.9** | **20.3** | **14.1** | **41.1** | **138.2** | **31.3** | 282.08 |

Table 4: The "in-domain" evaluation of our model and state-of-art ECE model on Flickr30K-EE test set. "Ref-Caps" denotes the initial quality of given reference captions.

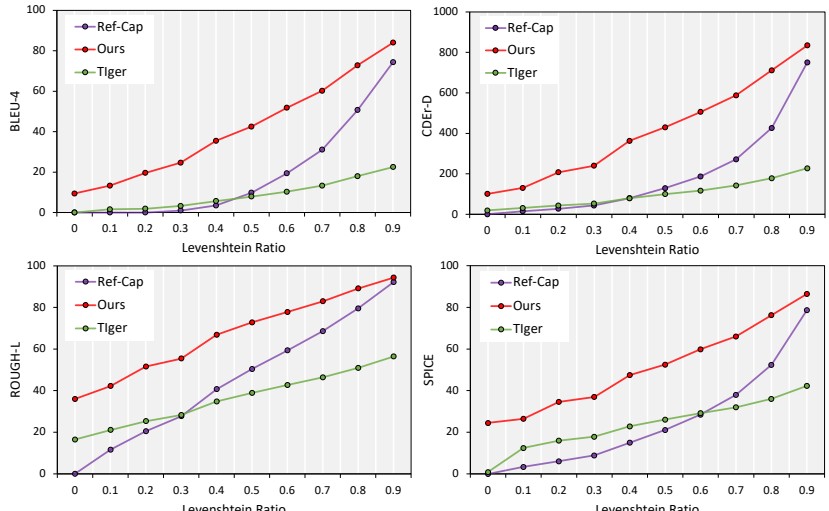

Figure 7: Performance of our model and state-of-the-art ECE models on out-of-domain GT-based reference captions. "Ref-Caps" denotes the quality of given GT-based reference captions.

**Setting A: GT-based Reference Captions.** The GT-based reference captions were constructed based on the GT-Caps in the Flickr30K-EE test set. We systematically replaced words in the GT-Caps with other random words from the vocabulary, resulting in the creation of various out-of-domain Ref-Caps. These Ref-Caps varied in terms of their Levenshtein ratio, ranging from 0.9 (*i.e.*, with only a few incorrect words) to 0 (*i.e.*, where all words were replaced with random ones). We evaluated edited captions against their single GT-Cap.

**Results.** As shown in Figure 7: Our model successfully improves the quality of all the GT-based Ref captions. In contrast, TIger struggles when editing Ref captions with either "minor" or "severe" errors, and even degrading the caption's quality (*e.g.*, Ref-Caps with a ratio larger than 0.4) by inadvertently removing accurate words or failing to introduce accurate details.

**Setting B: Zero-GT Reference Captions.** We constructed Zero-GT reference captions based on the Flickr30K test set. Specifically, each editing instance consists of a single image and a Ref-Cap with ten random words. Subsequently, we evaluated the edited captions against their corresponding five GT-Caps. All results are reported in Table 5.

**Results.** From Table 5 we can observe: 1) Given the image, two models achieve their best performance with five and six editing steps, respectively. Our DECap successfully edits the sentence with all random words into a reasonable caption. In contrast, TIger faces challenges in doing so. 2) For efficiency metrics, our DECap achieves significantly faster inference speed compared to TIger.

# E    ABLATIONS FOR THE NUMBER OF RANDOM WORDS

In this section, we run a set of ablation studies about the influence of different numbers of random words on caption generation. Specifically, similar to the setting in Sec. 4.3, we constructed input instances consisting of a single image from the COCO test set and a Ref caption with $n$ random words,

| Model | Unpaired Data | Step | Quality Evaluation | | | | | | | Inference Time(ms) |
|---|---|---|---|---|---|---|---|---|---|---|
| | | | B-1 | B-2 | B-3 | B-4 | R | C | S | |
| TIger | ✘ | 5 | 4.4 | 2.6 | 1.0 | 0.5 | 17.2 | 3.5 | 2.2 | 783.00 |
| **DECap** | ✔ | 6 | **70.1** | **47.9** | **29.8** | **17.5** | **46.7** | **45.7** | **13.3** | 427.00 |

Table 5: Performance of our model and state-of-the-art ECE model on Zero-GT reference captions.

| Model | Random Words | Step | B-1 | B-2 | B-3 | B-4 | M | R | C | S |
|---|---|---|---|---|---|---|---|---|---|---|
| **DECap** | 8 | 10 | 75.7 | 59.3 | 44.5 | 32.4 | 26.3 | 56.5 | 109.7 | 20.1 |
| | 9 | 10 | **80.2** | **63.3** | **47.9** | **35.5** | 27.8 | 58.0 | 118.1 | 21.5 |
| | 10 | 10 | 78.0 | 61.4 | 46.4 | 34.5 | 28.6 | **58.0** | **119.0** | 21.9 |
| | 11 | 10 | 75.9 | 58.8 | 44.3 | 32.9 | 28.9 | 57.1 | 115.7 | 22.4 |
| | 12 | 10 | 72.0 | 56.3 | 42.2 | 31.2 | **29.0** | 56.1 | 109.3 | **22.7** |

Table 6: Performance of our model on the COCO with different numbers of input random words.

where $n \in \{8, 9, 10, 11, 12\}$. The edited captions were then evaluated against their corresponding five ground-truth captions.

From Table 6 we can observe: 1) DECap's performance consistently improves as the number of random words increases from 8 to 10 and then starts to decline beyond 10 random words. 2) Given that the average length of ground-truth captions in COCO is around 10 words, DECap achieves its highest CIDEr-D score when editing sentences with 10 random words. While BLEU-N metrics tend to favor shorter sentences, DECap obtains the best BLEU scores with competitive CIDEr-D scores when editing sentences with 9 random words. Additionally, as the number of random words increases, DECap generates more semantic information about the image, including objects and attributes, resulting in higher SPICE scores. However, this increase in semantic content can also lead to issues like repetition and the introduction of extraneous details, referring to objects or information present in the image but not explicitly mentioned in the ground-truth captions. This can all potentially lead to a decline in the quality evaluation of the generated captions. 3) Based on these findings, we select 10 words as a balanced choice for caption generation.

## F ABLATIONS FOR THE DISTRIBUTION OF EDIT TYPES

As discussed in Sec.3.2, the distribution over edit types plays a crucial role in balancing different noising operations and training diverse denoising abilities. Therefore, in this section, we conduct a series of ablation experiments to examine the impact of varying distribution settings for the edit types within the edit-based noising process. Specifically, the probabilities for the noising edit operations REPLACE, DELTE, and INSERT is denoted as $\alpha$, $\beta$, and $\gamma$, respectively. While these probabilities are parameterized by several factors, such as the current state of the caption and the length of the ground-truth caption, we perform ablations by imposing global control over these probabilities. For instance, we explore settings where $\alpha=\beta=\gamma$, $\alpha>\beta=\gamma$ and $\beta=\gamma=0$.

**Setting.** We train the DECap on the COCO training set with different distributions of edit types with the same diffusion set $T = 10$. During testing, we constructed input instances consisting of a single image from the COCO test set and a Ref-Cap with ten random words. The edited captions were then evaluated against their corresponding five GT-Caps.

**Results.** From Table 7, we can observe: 1) In comparison to the even distribution of edit types, where $\alpha=\beta=\gamma$, DECap demonstrates improved performance when we emphasize the denoising ability of the replacement operation with the distribution $\alpha>\beta=\gamma$. This suggests that the replacement operation is more flexible and efficient in correcting words than the sequence operation of first deletion and then insertion. 2) When we trained DECap with an exclusive focus on the replacement operation and omitted the deletion and insertion abilities, setting $\beta=\gamma=0$, there is a noticeable decline in the quality of generated captions. This indicates that DECap's ability to adjust caption length by adding more description or removing repetitions is compromised. 3) These results suggest that the distribution with $\alpha>\beta=\gamma$ could be a sensible choice for caption generation. Importantly, our method allows for flexible adaptation, enabling us to set different edit type distributions tailored to specific tasks or requirements.

| Model | Distribution of Edit Types | B-1 | B-2 | B-3 | B-4 | M | R | C | S |
|-------|---------------------------|-----|-----|-----|-----|---|---|---|---|
| **DECap** | $\alpha = \beta = \gamma$ | 77.4 | 60.8 | 45.8 | 34.0 | 28.6 | 57.8 | 117.4 | 21.8 |
| | $\alpha > \beta = \gamma$ | **78.0** | **61.4** | **46.4** | **34.5** | **28.6** | **58.0** | **119.0** | **21.9** |
| | $\beta = \gamma = 0$ | 77.3 | 60.6 | 45.7 | 33.8 | 25.8 | 57.8 | 116.6 | 21.7 |

Table 7: Performance of our model on the COCO with different distributions of noising edit types.

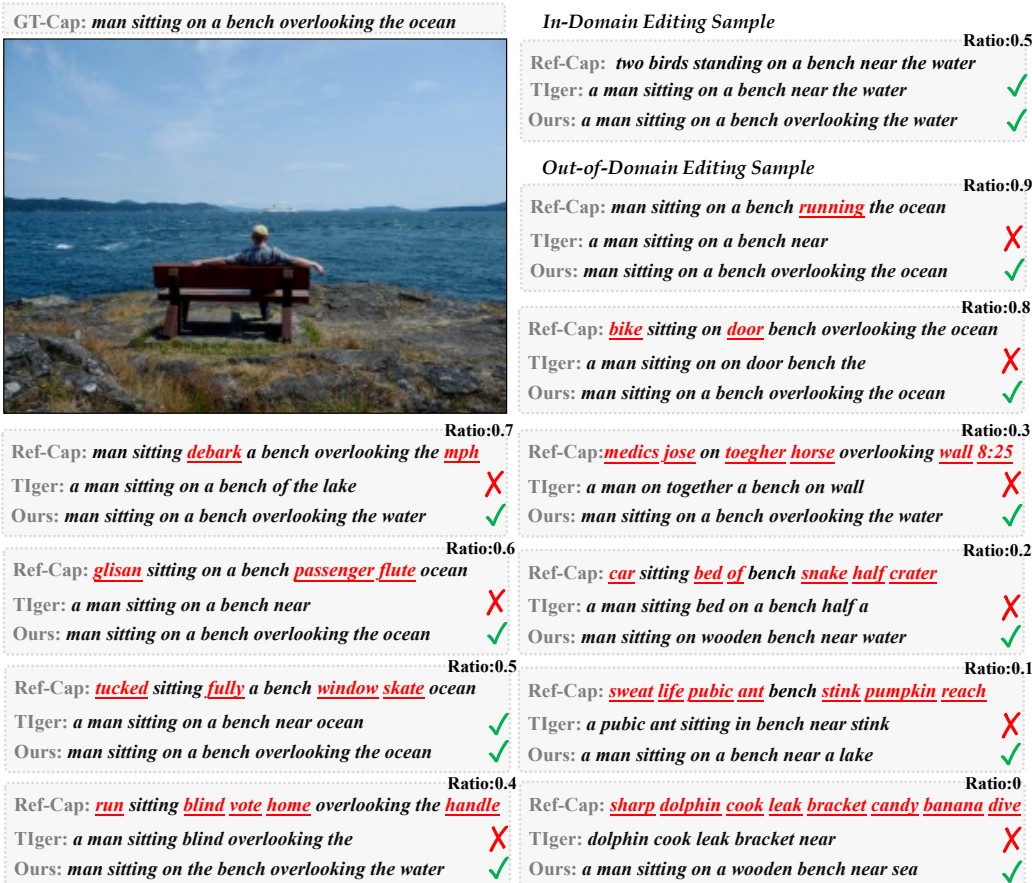

Figure 8: Editing results of existing ECE model and our DECap. Specifically, the "*In-Domain*" editing sample denotes the carefully selected Ref-Cap from the COCO-EE test set, while the "*Out-of-Domain*" editing sample denotes the GT-based reference captions constructed by directly replacing the GT-Cap with random words.

## G   MORE VISUALIZATION RESULTS.

**Generalization Ability**. As illustrated in Figure 8, existing ECE model TIger (Wang et al., 2022) exhibits limited capability in refining *Out-of-Domain* editing samples, particularly when the similarity between Ref-Cap and GT-Cap deviates significantly from the balanced value (*i.e.*, Ratio 0.5) seen in training data. In contrast, DECap displays a remarkable generalization ability, successfully editing both *In-Domain* and *Out-of-Domain* samples with diverse editing scenarios, covering ratios ranging from 0.1 to 0.9. Moreover, DECap is also capable of editing an entirely random word sequence (*i.e.*, Ratio 0), producing high-quality captions in such challenging scenarios.

**Potential in Controllability**. As illustrated in Figure 9, our model effectively edits sentences based on specific input control words. All the generated captions maintain the order of the provided control words, ensuring both fluency and semantic relevance to these control words.

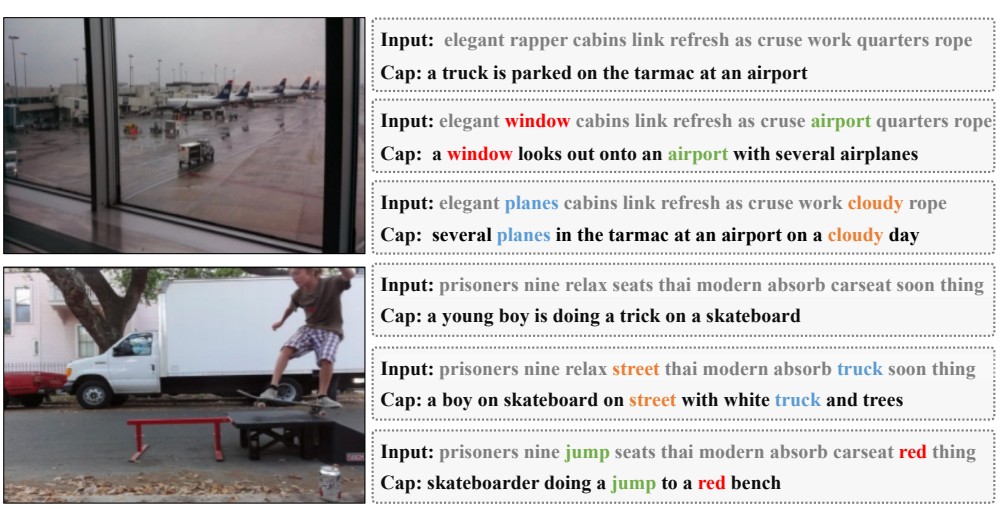

Figure 9: Controllability of DECap. The grey words represent random words from the vocabulary, other colored words represent the manually placed control words.

