# OpenReview forum: "DECap: Towards Generalized Explicit Caption Editing via Diffusion Mechanism"
_ICLR.cc/2024/Conference — ICLR 2024 Conference Withdrawn Submission_

### Official Review · Reviewer_x69y · 2023-10-31

**Soundness:** 3 good
**Presentation:** 3 good
**Contribution:** 2 fair
**Rating:** 5
**Confidence:** 4

**Summary:**

This paper focuses on the generalization ability of the existing explicit caption editing methods. To this end, they propose a new diffusion-based explicit caption editing method named DECap. With the edit-based noising and denoising processes, DECap performs a strong generalization ability in various caption editing scenarios.

**Strengths:**

1. **The proposed method is effective and well-designed.** The proposed diffusion-based explicit caption editing is natural and well-designed.
2. **The motivation is clear.** The generalization of the ECE task is an important problem.
3. **The main results are strong**. The experiments on Tab.1 and Tab.2 show that the DECap achieves better caption performance with faster inference time.

**Weaknesses:**

1. **The meaning of the ECE task.** The most significant weakness of this paper is the meaning of the explicit caption editing task. If we use GPT-4V to edit the caption with the image, will show great generalization to the out-of-domain samples? Can you provide a comparison with the LMM models (GPT-4V, LLava, MiniGPT-v2, Openflamingo) for explicit caption editing?

2. **The metrics are not enough.** I think the CLIP-Score and human evaluation of Tab.1 and Tab.2 is necessary.

3. **The Section 4.2.2 Out-of-domain evaluation is not convinced enough.**  The settings of A and B are not enough to get the conclusion that your method has better generalization. You should select an out-of-domain dataset to evaluate the method's generalization. Just having better BLEU-4/CIDEr-D/ROUGE-4/SPICE is not enough.

4. **The reproducibility of the models.** It seems that the proposal model DECAP is a well-designed model. I am not sure if the community can reproduce the DECAP without the provided code.

**Questions:**

As shown in weaknesses.

---

### Official Review · Reviewer_Y6nt · 2023-11-01

**Soundness:** 2 fair
**Presentation:** 3 good
**Contribution:** 2 fair
**Rating:** 5
**Confidence:** 4

**Summary:**

This paper propose a new Diffusion-based Explicit Caption editing method for explicit caption editing (ECE), which refine the reference image captions through a sequence of explicit edit operations.

**Strengths:**

This paper is well written and easy to follow.

The proposed method formulate the ECE task as a denoising process, and the noising process helps the eliminate the need for meticulous paired data selection by directly introducing word-level noises for model training.

The model is evaluated with extensive experiments under both in-domain and out-of-domain datasets and demonstrate superious performance when compared with the state-of-the-art ECE model TIger.

**Weaknesses:**

1. The motivation of Explicit Caption Editing is unclear, especially under the out-of-domain setting. Table 2 represents the results of ECE models with Zero-GT reference captions (captions with ten random words). However, in real-world scenarios, we typically don't encounter captions with random words. Moreover, most conventional captioning models are essentially zero-reference models capable of producing precise captions.

2. The results of DECap in Table 2 (row 2) and Table 3 (row 13) is not the same. From the paper I found that the models are both trained with 10 Steps and tested with Ref-Cap with ten random words on COCO test set. What is the difference between the two models?

3. DECap is not out-performing the state-of-the-art ordinary image captioning models. For instance, the CIDEr of DECap is 121.2 on COCO test set, while the state-or-the-art results are 134.2[1r], 127.4[2r].

[1r] DIFNet: Boosting Visual Information Flow for Image Captioning, CVPR 2022.
[2r] Comprehending and Ordering Semantics for Image Captioning	, CVPR 2022.

**Questions:**

See Weaknesses. I am looking forward to the author responses and willing to adjust my rating if my concern are adequatly addressed.

---

### Official Review · Reviewer_4DTL · 2023-11-01

**Soundness:** 3 good
**Presentation:** 3 good
**Contribution:** 3 good
**Rating:** 5
**Confidence:** 3

**Summary:**

The proposed methods solve ECE task that edits caption to be more human-readable using a diffusion model. In the forward process, they gradually edit words, while in the backward process, they employ a network to predict the editing operations and the words they were replaced with.

**Strengths:**

In contrast to conventional captioning models, the proposed approach offers a degree of controllability based on the given input. Also, it demonstrates a good robustness in effectively correcting even when incorrect words are introduced.

**Weaknesses:**

Since the model predicts multiple time steps, it is expected to be more time-consuming. There may be doubts about the utility of such an editing approach compared to one-shot captioning models. For example, models like SCD-Net seem to perform better on COCO, and this raises questions about the effectiveness of the editing method in this specific task.

**Questions:**

It is wondering if the results in the captions vary significantly when provided with completely random inputs. It would also be interesting to see if there are advantages demonstrated in results compared to generating captions from scratch, starting with no input at all.

---

### Official Review · Reviewer_dhFv · 2023-11-06

**Soundness:** 1 poor
**Presentation:** 3 good
**Contribution:** 2 fair
**Rating:** 3
**Confidence:** 3

**Summary:**

This paper focuses on the problem of explicit caption editing (ECE). The author tackled the ECE problem from the novel perspective of discrete diffusion, where the noise is added to GT caption step-by-step, and the model learns to denoise the noised caption. The proposed method demonstrates strong performance in in-domain benchmarks and better generalizability in out-domain benchmarks compared to previous SoTA TIger.

**Strengths:**

- strong performance for in-domain benchmarks
- good generalizability for out-domain benchmarks
- efficient inference

**Weaknesses:**

This is not a fair comparison with the previous method, TIger, under the ECE setting. In the ECE setting paper [1], this setting is constructed with the following criteria:

1. $\textbf{Human Annotated Captions.}$ Both Ref-Cap and GT-Cap should be written by humans to avoid grammatical errors.
1. $\textbf{Image-Caption Similarity.}$ The scene described by the Ref-Cap should be similar to the scene in the image.
1. $\textbf{Caption Similarity.}$ Paired Ref-Cap and GT-Cap should have a certain degree of overlap and similar caption structure to avoid completely regenerating the whole sentence or roughly breaking the structure of Ref-Cap.

Therefore, the constructed dataset has similar Ref-Cap (input captions that need correction) and GT-Cap (ground-truth caption). For example, Ref-Cap "Motorcyclists are stopped at a stop sign" and GT-Cap "Motorcyclists are in a close race around a corner". It is not surprising that a model trained on such a dataset generalized poorly to the contrived out-domain data with randomly replaced and dropped words and usually grammatically wrong Ref-Cap in this paper.

Furthermore, the proposed nosing process can be viewed as an aggressive data augmentation method. It is unclear whether the performance improvement, either for in-domain or out-domain, is actually coming from the discrete diffusion mechanism, the model design; or simply from the data augmentation.

To validate the claims the author made in this paper, it is critical to ablate the contribution of data augmentation. For example, the author may consider training TIger with their synthesized noised data to make sure both models see the same amount of data and then compare the performance. If in this ablation TIger and DECap perform roughly the same, then the major contribution is limited to proposing a data augmentation strategy, which is not significant enough from the reviewer's point of view.

[1] Wang, Zhen, et al. "Explicit image caption editing." European Conference on Computer Vision. Cham: Springer Nature Switzerland, 2022.

**Questions:**

Please see the weakness section. The reviewer's biggest concern is that performance improvement may mainly come from data augmentation.